# UniEEG: Advancing Universal EEG Representation with Electrode-Wise Time-Frequency Pretraining

## Abstract

Previous electroencephalogram (EEG) models typically exhibit limited performance and generalization by collecting data specifically for targeted EEG tasks. Recognizing this limitation, we propose UniEEG, the first electrode-wise time-frequency pretraining model, designed to overcome barriers across diverse tasks and data in EEG modeling. We collect data from nearly 20 publicly available EEG datasets, including 6 EEG tasks, significantly extending the data volume. The collected EEG data are standardized and split to individual electrodes as the input of UniEEG, enabling full compatibility with diverse EEG data from different acquisition devices and task paradigms. Meanwhile, leveraging a time-frequency transform method, UniEEG adeptly processes EEG signals characterized by signal noises and time delays. In the training phase, we employ an encoder-decoder architecture and a mask signal modeling strategy on time-frequency dimension, learning the electrode-wise universal EEG representation. In the fine-tuning phase, multi-electrode EEG signals from various tasks are consolidated into individual electrodes. The predictions for downstream tasks are then obtained through the pre-trained encoder and an additional prediction module. Furthermore, the proposed UniEEG achieves state-of-the-art performance across different EEG tasks, demonstrating an amazing ability to universal EEG feature representation. Code, data and models would be available upon acceptance.

## 1 Introduction

## 2 Introduction

Electroencephalogram (EEG) signals are recorded by placing multiple electrodes at different locations on the scalp, capturing temporal fluctuations in voltage that reflect underlying brain activity. EEG has the advantages of non-invasive, multi-channel recording and high temporal resolution, and has been applied in many fields such as brain computer interface Wang et al. (2006); Li et al. (2012); Zhang et al. (2015), cognition Li et al. (2016), sentiment analysis Koelstra et al. (2012); Zheng & Lu (2015), motor imagery Cho et al. (2017); Schalk et al. (2004) and so on. With the development of deep learning, EEG processing methodology is evolved to CNN Cecotti & Graeser (2008), RNN Tsiouris et al. (2018), Transformer Sun et al. (2021b); Xie et al. (2022a) methods, etc. Meanwhile, the recent success of pre-training models on natural language processing Devlin et al. (2018); Radford et al. (2018); Touvron et al. (2023) and computer visionRadford et al. (2021); He et al. (2022); Oquab et al. (2023); Kirillov et al. (2023); Li et al. (2023b); Liu et al. (2023); Zhang et al. (2023) , which capture a universal representation with large-scale unlabeled data and the representation can be adapted to various downstream tasks, inspires the emergence of EEG pretraining models, which would hopefully revolutionize the brain-interface field and community.

However, the construction of EEG pretraining models continues to face challenges. The challenges can be summarized as following:

**1) Limited Data Availability.** EEG data collection is challenging, requiring specialized equipment and expertise. Annotating and segmenting data is time-consuming, resulting in small labeled datasets for specific tasks. The scarcity of labeled data hinders the training of effective pretraining

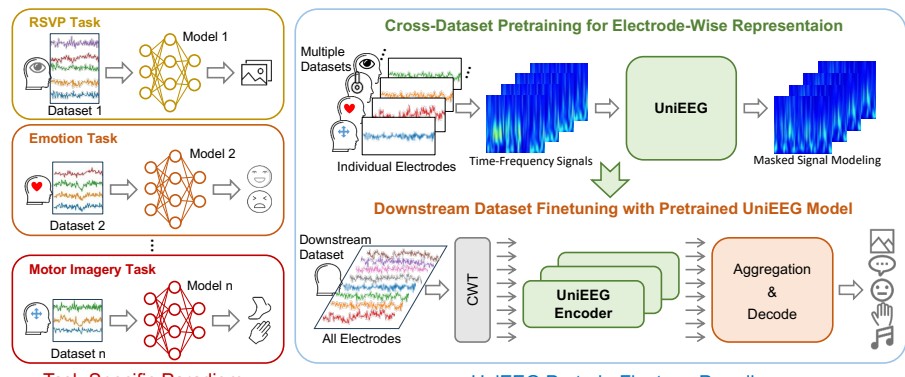

Figure 1: **Comparison between previous training paradigm and UniEEG**. Compared to previous task-specific EEG paradigms focus on single dataset or task, UniEEG adopts cross-dataset electrode-wise pretraining to extend data volume and enhance universal representation. For specific tasks, UniEEG finetunes its pretrained encoder to the particular dataset, offering a more versatile and efficient approach to EEG data analysis.

models, limiting their generalization. Therefore, it is necessary to explore strategies for utilizing large-scale unlabeled EEG data, potentially incorporating semi-supervised or unsupervised learning methods.

**2) Diverse EEG Data Configurations.** Different EEG acquisition setups, electrode configurations, and experimental paradigms lead to diverse data formats. Handling varied EEG data formats is crucial for compatibility with the pretraining models. Therefore, it is important to standardize or preprocess diverse EEG data formats or unify EEG experimental paradigms, ensuring consistency in input units for effective pretraining.

**3) Ineffective Representation Learning Paradigms.** EEG signals often exhibit a low signal-to-noise ratio (SNR), and various noise types pose challenges in representation learning. Current representation learning paradigms(CNN, RNN, and Transformer) may face challenges in addressing diverse EEG characteristics effectively. How to adjust these learning paradigms for EEG data, capturing effectively information and reducing the influence of low SNR and diverse noise types, needs further consideration.

Therefore, the key to establishing an effective EEG pre-training model lies in designing a sensible data format and learning paradigm that is **"Universal" on data, tasks and paradigms.**

To address these challenges, we propose **UniEEG**, the first electrode-wise time-frequency pretraining model, aiming to fully leverage the existing EEG data and generate a universal EEG representation from time-frequency EEG signals. We introduce the following strategies: **1) Extend the data volume.** The acquisition of EEG data is both costly and intricate, making it impractical for researchers to amass extensive pretraining datasets. Despite the relatively modest scale of individual tasks within the disclosed EEG data, the cumulative dataset size is substantial, aligning well with the requirements for pretraining at scale. Therefore, we gathered an extensive array of publicly available EEG datasets (18 EEG datasets on 6 tasks), effectively augmenting the overall data volume (about 2M samples). **2) Standardize diverse EEG data formats.** Although the experimental paradigms show significant differences, the basic unit of the EEG signal is the electrode. Therefore, we explore the feasibility of employing a single electrode as the input for our model to overcome the challenge of non-generic data across different experimental paradigms. This approach dismantles the non-generic barrier between EEG signals of different paradigms. **3) Construct effective representation learning paradigms.** Since EEG has the characteristics of low signal-to-noise ratio, large randomness and time delay, we believe that simple temporal EEG signals are not enough for feature extraction and semantic analysis. In this paper, we exploit time-frequency analysis methods like continuous wavelet transform (CWT) to obtain time-frequency features of EEG, as the input of the model. We introduce an encoder-decoder architecture to extract semantic information and reconstruct the time-frequency EEG, learning the universal EEG representation with self-supervised

paradigm. Following MAE He et al. (2022), we pretrain the UniEEG with masked signal modeling strategy for learning effective feature representation.

To summarize, our contributions are as follows:

- We introduce UniEEG, the pioneering electrode-wise time-frequency pre-training model for EEG signals, which focuses on capturing the universal representational of EEG signals and serves as a valuable pretraining model for a spectrum of downstream EEG tasks.

- We present an expanded EEG dataset that gathers data from nearly 20 publicly available EEG datasets. The dataset standardizes EEG into an electrode-wise time-frequency representation, addressing compatibility challenges across EEG data and tasks during pretraining.

- We design an encoder-decoder architecture and a mask signal modeling strategy on time-frequency dimension, learning the electrode-wise universal EEG representation.

- We conducted a thorough and systematic study of EEG pre-training and downstream tasks. The proposed UniEEG significantly improves the performance on various EEG tasks and shows a strong ability on universal EEG feature representation.

## 3 RELATED WORK

### 3.1 EEG CLASSIFICATION

The end-to-end EEG classification Li et al. (2019); Song et al. (2018); Ding et al. (2022); Li et al. (2022); Altaheri et al. (2022); Du et al. (2023); Zhang et al. (2022); Li et al. (2023a); Yang et al. (2023); Tabar & Halici (2016); Yao et al. (2018); Bashivan et al. (2015) aims to directly processes raw EEG data to perform a specific classification task, where the labels are usually defined as the category of the stimula, like motor imagery or image-based EEG classification.

Schirrmeister et al. Schirrmeister et al. (2017a) attempt to exploit CNN and propose Shallow ConvNet, Deep ConvNet, and Hybrid ConvNet to encode the EEG signal for classification. To fully leverage the spatial domain correlations within EEG signal channels, Sun et al. Sun et al. (2021a) establish a trainable adaptive matrix and introduce adaptive spatio-temporal graph convolutional networks (ASTGCN). Ingolfsson et al. Ingolfsson et al. (2020) propose EEG-TCNet to further utilizes depthwise convolution and separable convolution techniques to embed the signal, gaining promising results. Li et al. Li et al. (2020) employ the methodology of attention mechanism and propose a multi-scale fusion convolutional neural network (MS-AMF). Furthermore, Fan et al. Fan et al. (2021) introduce a newly designed attention module (3D-AM) to automatically learn the importance of different electrodes, time points, and feature maps. Most recently, Luo et al. Luo et al. (2023) propose a dual-branch spatio-Temporal-Spectral transformer, which concurrently extracts distinctive features from EEG signals in both the spatial-temporal and spectral-temporal domains. The works Yao et al. (2018); Bashivan et al. (2015) further introduce antoencoders to model the EEG representation.

The previous arts are well-designed architecture and achieve promising results for specific tasks. However, the specific architecture makes it difficult to generalize in other paradigms. A unified architecture is required to create a universal representation for EEG signals, which is the main focus of our work.

### 3.2 MASKED SIGNAL MODELING

Masked Signal Modeling (MSM) has recently achieved great success in natural language processing Devlin et al. (2018) and computer vision He et al. (2022); Xie et al. (2022b); Wei et al. (2022). It functions as a generalized denoising autoencoder, which reconstructs the original data from a portion of the input sentence. For example, Bert Devlin et al. (2018) proposes to mask and predict the language words and MAE He et al. (2022) proposes to mask and reconstruct the image patches. Most close to our work is SC-MBM Chen et al. (2023), which introduces sparse-coded masked brain modeling to mask and construct the fMRI data. However, there are no evidence to validate the effectiveness of MSM in EEG signal, which is the main focus of our work.

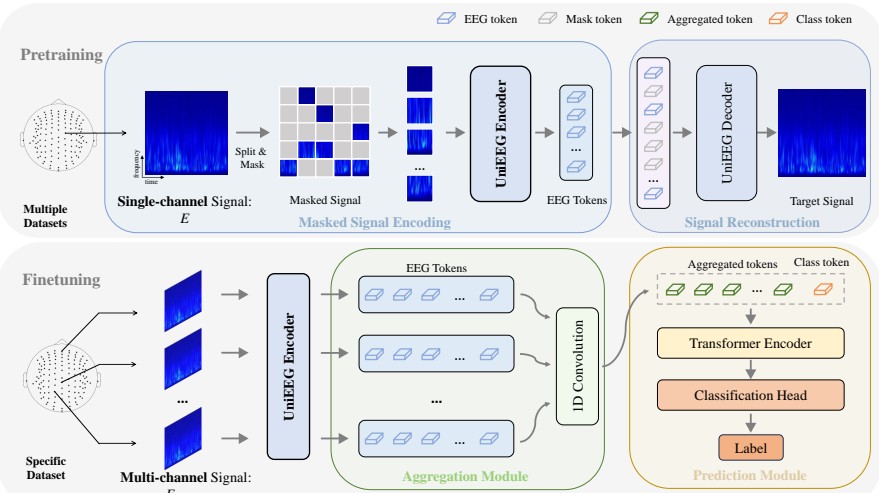

Figure 2: The overall architecture of UniEEG. (1) **Pretraining**. UniEEG consists of two components: a UniEEG Encoder that maps the observed time-frequency signal to a latent representation and a UniEEG Decoder that reconstructs the original signal from the latent representation. In the UniEEG Encoder, mask signal modeling strategy is employed on time-frequency dimension. A subset of observed single-channel signals without mask tokens pass the encoder while the UniEEG Decoder reconstructs the original signal from the latent representations and additional mask tokens. (2) **Finetuning**. We first extract signal features for each channel and then perform aggregation (1D convolution) along the EEG channel dimension to fuse the different channels. A classification head is then employed to get the predictions, which takes the flattened convolutional features as input. Note that the pretraining process is executed with a single EEG channel, while during finetuning period all of the channels should be utilized.

## 3.3 PRETRAINING MODELS

Benefiting from the great performance and generalization of large pretraining models, the natural language processing tasks Devlin et al. (2018); Radford et al. (2018); Touvron et al. (2023) and computer vision tasks Oquab et al. (2023); Kirillov et al. (2023); Radford et al. (2021); Li et al. (2023b); Liu et al. (2023); Zhang et al. (2023) have achieved a great boost in recent years. These pretraining methods, which are usually based on transformer Vaswani et al. (2017), enhance the reasoning ability of models to a large extent with the utilization large-scale data and model capacity. Inspired by them, the proposed UniEEG pretrains the EEG model with large-scale data to generate a universal representation. We hope that such cross-time, cross-space and cross-disciplinary EEG pretraining model could have an important research value and significance for the study and analysis of EEG signals.

## 4 METHOD

In this paper, we propose UniEEG, the first electrode-wise EEG pretraining model for universal time-frequence representation. To implement the proposed method, we collect and preprocess over 20 EEG datasets to construct a large-scale universal single electrode time-frequency EEG dataset.

### 4.1 PRETRAINING DATA COLLECTION AND PREPROCESS

#### 4.1.1 EEG DATA COLLECTION

To prepare our model for EEG data analysis, we have gathered numerous publicly accessible EEG datasets and transformed them into a time-frequency format. Our universal EEG dataset comprises 18 EEG datasets that cover 6 tasks, which include: 1) *Sentiment Analysis* Koelstra et al. (2012); Zheng & Lu (2015): using EEG data to identify and evaluate the emotional state of an individual; 2) *Music Imagery* Daly et al. (2019): studying and analyzing the electrical activity of the brain while

a person imagines or mentally processes music; 3) *Event-Related Potential (ERP)* Chavarriaga & Millán (2010): analyzing the brain's electrical activity in response to specific events or stimuli, such as visual, auditory, or sensory stimuli. 4) *Motor Imagery*Brunner et al. (2008); Steyrl et al. (2014); Leeb et al. (2008); Cho et al. (2017); Schalk et al. (2004); Luciw et al. (2014); Kaya et al. (2018); Schirrmeister et al. (2017b); Bhatt (2012); Dornhege et al. (2004): referring to the mental simulation or visualization of specific motor movements or actions without physically performing them. 5) *Image-based EEG Classification* Gifford et al. (2022); Grootswagers et al. (2022); Spampinato et al. (2017): using EEG data to classify images or other visual stimuli. 6) *Speech Imagery Classification* Nguyen et al. (2017): using EEG data to categorize or classify different aspects of speech without physically hearing them. More information on these tasks can be found in the Appendix.

### 4.1.2 DATA PREPROCESS

The primary challenge in preprocessing large-scale EEG signals is the variation in collection parameters (sampling frequency and numbers of electrodes) on different datasets with different collection paradigms.

First, the time-domain EEG signals are transformed into the time-frequency domain using Continuous Wavelet Transform (CWT). Then we apply simple filtering to the signal by removing frequencies below 2Hz or above 50Hz.

During pre-training, we should consider the differences in channel numbers (the number of electrodes) and data length (collection time) for different collection paradigms. Previous works Alotaiby et al. (2015); Jiao et al. (2020) have found that there are commonalities between the EEG representations of multiple channels caused by the signal acquisition principle. These representations could be modelled in a similar way even if the channels are different. Thus we split each channel of EEG data and treat them as independent samples. For data length, following Jiao et al. (2020), we have a crop step before resizing. We first randomly crop the input signal to a random duration along the time dimension and then resize it. In this way, the model could see input signals with flexible length, which could achieve better results on data of most lengths It should be mentioned that when performing downstream tasks, the data in different channels are not divided but aggregated by a fusion operation, and the data length is only resized to the preset dimension in the pretraining period.

Further, considering the variation of sampling rate in different datasets, we re-sample the raw EEG data to 100Hz, where we employ linear interpolation for upsampling and uniform sampling for downsampling. Considering the information redundancy in time-series signal, the data loss caused by the sampling is acceptable on semantic analysis. Moreover, we exploit time-frequency signal as input, which introduces additional frequency information and compensates for this loss.

### 4.2 ARCHITECTURE AND PRETRAINING OF UNIEEG

To capture the universal representation for EEG signals, we propose UniEEG, an electrode-wise time-frequency pretraining model, which aims to capture the universal EEG representations in spite of various stimuli.

The general architecture of our proposed UniEEG is shown in Fig.2, which consists of two components: a UniEEG Encoder that maps the observed time-frequency signal to a latent representation and a UniEEG Decoder that reconstructs the original signal from the latent representation. Following He et al. (2022), the UniEEG Encoder is designed to operate only on a subset of observed single-channel signals without mask tokens while the UniEEG Decoder constructs the original signal from the latent representations and additional mask tokens.

After preprocessing, a time-frequency signal $E$ has size of $F \times S \times 1$, where $F$ represents the frequency range and $S$ represents the number of sampling points. As a placeholder, the last dimension 1 is set to match visual images. In this section, the signal is treated as an image, where the pixel value in $(h, w)$ represents the energy value of the signal in frequency $h \in F$ and sampling point $w \in S$.

Table 1: Comparisons with SOTA. We show the performance on different datasets for different tasks. From left to right, the tasks are: sentiment analysis (SA), motor imagery (MI), image-based EEG classification (IEC) and speech-based EEG classification (SEC). Note that we only report the results with only EEG as the input and only report the holdout validation results (compared with leave-one-subject-out validation) for fairness.

| Method | SA | | | MI | | | | | | IEC | | SEC |
|---|---|---|---|---|---|---|---|---|---|---|---|---|
| | SEED Zheng & Lu (2015) | Neuro-Marketing Yadava et al. (2017) | DEAP Koelstra et al. (2012) | MIBCI Cho et al. (2017) | Grasp and Lift Kaggle (2021) | Motor Imagery Kaya et al. (2018) | BCI III 4A Dornhege et al. (2004) | BCI IV 2A Brunner et al. (2008) | BCI IV 2B Leeb et al. (2008) | Ger+ Aus Gifford et al. (2022) | EEG-Based Visual Grootswagers et al. (2022) Spampinato et al. (2017) | Speech Imagery Nguyen et al. (2017) |
| SOTA | 93.46Gupta et al. (2019) | 70.0Yadava et al. (2017) | 90.7Bazgir et al. (2018) | - | 98.1Kaggle (2021) | - | 74.28Dornhege et al. (2004) | 78.82Temiyasathit et al. (2014) | 78.93Lee & Choi (2018) | - | 82.95Spampinato et al. (2017) | - |
| Image-wise AEYao et al. (2018) | 84.21 | 70.59 | 81.34 | 76.45 | 89.29 | 49.89 | 63.19 | 75.17 | 77.34 | 17.69 | 84.38 | 49.70 |
| ConvNetBashivan et al. (2015) | 86.30 | 76.82 | 80.4 | 69.2 | 90.03 | 49.84 | 64.32 | 69.74 | 82.10 | 20.43 | 85.23 | 55.20 |
| Ours w/o pretraining | 91.76 | 81.95 | 79.34 | 68.94 | 98.21 | 46.8 | 75.01 | 80.74 | 81.69 | 18.48 | 83.16 | 56.47 |
| Ours | 93.85 | 83.71 | 92.88 | 79.63 | 98.5 | 59.20 | 78.64 | 82.35 | 82.26 | 22.59 | 84.53 | 59.78 |

### 4.2.1 UNiEEG ENCODER

We first divide $E$ into regular non-overlapping patches. Then we randomly sample the patches in a percentage of $R$ and mask the remaining ones, which are subsequently embedded by a linear projection layer with added positional embeddings. Just as in a standard MAE, the masked patches are removed and no mask tokens are used, which enable the expansion of encoder with limited cost of compute and memory. The embedded signal patches are then passed as input to self-attention layers, resulting in the latent representations of EEG signals.

### 4.2.2 UNiEEG DECODER

We then exploit a UniEEG Decoder to reconstruct the original signal from the encoded unmasked signal patches and added mask tokens. Inspired from Devlin et al. (2018), the mask token is a learnable embedding with the same size as the encoded signal patch, which indicates the place where the original patch has been masked and removed. The encoded unmasked patches are placed in their original location in the whole signal. These masked and unmasked patches, added with positional embeddings, are passed as input to another attention layers to generate the original signal. The reconstruction targets of UniEEG Decoder are the pixel values of each masked patch.

The overall loss of the pretraining process is the mean squared error of pixel values between the reconstructed signal image and original signal image on masked patches.

## 4.3 FINETUNING UNiEEG ON DOWNSTREAM TASKS

UniEEG is conducted in a self-supervised way on the universal EEG datasets. To evaluate the capability of proposed representation, we perform extensive experiments on diverse down-streaming tasks by finetuning the pretrained UniEEG Encoder. As illustrated in Fig 2, the pretraining process is executed for every individual EEG channel, while during finetuning period all of the channels should be used.

Specifically, for an EEG signal $E_{\{1,...,C\}}$ with $C$ channels, we first extract signal features for each channel, resulting features with size of $C \times P_H \times P_W \times D$, where $P_W$, $P_H$ represent the patch size in height and width and $D$ is the hidden dimension. We perform 1D convolution along the EEG channel dimension to fuse the different channels. We then apply a classification head to the flattened convolutional features to get the predictions.

## 5 EXPERIMENT

### 5.1 EXPERIMENTAL SETUP

#### 5.1.1 PRETRAINING AND EVALUATION DATASETS

To ensure the diversity of the pretraining data in UniEEG, we gather a comprehensive collection of 18 publicly available EEG datasets. During pretraining phase, UniEEG leverages a mixed dataset compiled from 16 of these datasets, ensuring a wide range of EEG patterns are encompassed. For the finetuning phase, we select 12 datasets to evaluate the performance of UniEEG, all of which are oriented towards classification tasks. It's important to note that during pretraining, only the training sets are utilized. And during finetuning we report the results on the test sets of the selected 12 classification datasets.

Table 2: Comparison on decoder depth.

| Depth | Finetuning | | Frozen | |
|---|---|---|---|---|
| | Ger+Aus | Deap | Ger+Aus | Deap |
| 1 | 20.81% | 77.90% | 12.16% | 71.22% |
| 2 | 20.76% | 80.79% | 16.69% | 76.30% |
| 4 | 21.03% | 84.65% | 17.57% | 75.69% |
| 8 | **22.59%** | **85.61%** | **19.78%** | **79.79%** |
| 12 | 21.14% | 83.06% | 19.13% | 75.10% |

Table 3: Comparison on decoder width.

| Width | Finetuning | | Frozen | |
|---|---|---|---|---|
| | Ger+Aus | DEAP | Ger+Aus | DEAP |
| 128 | 21.26% | 82.46% | 17.34% | 75.45% |
| 256 | 20.32% | 85.26% | 18.90% | 74.08% |
| 512 | 21.48% | 84.37% | 19.56% | 76.64% |
| 768 | **22.59%** | **85.61%** | 19.78% | **76.79%** |
| 1024 | 21.35% | 83.18% | **19.85%** | 75.31% |

### 5.1.2 TRAINING DETAILS

UniEEG is trained using the PyTorch framework on 8 NVIDIA A100. The UniEEG encoder are initialized from MAE He et al. (2022). The initial learning rate is 0.0001 during pretraining and 0.0007 during finetuning. We utilize AdamW optimizer and adopt a warm-up learning rate during the training process. The whole pretraining for 10 epoches takes about 20 hours.

### 5.2 MAIN RESULTS

We evaluate the performance of the proposed UniEEG on four downstream tasks: sentiment analysis (SA), motor imagery (MI), image-based EEG classification (IEC) and speech-based EEG classification (SEC) (see Sec. B in detail), which are basic EEG tasks to learn the brain activities. Tab. 1 shows detailed comparisons on the 12 datasets, including several datasets that contain only data, but no profile results. Despite the diversity of the above tasks and datasets, our proposed UniEEG can obtain a universal EEG representation and has strong cross-task semantic analysis ability, achieving state-of-the-art performance across datasets.

Generally, UniEEG outperforms most previous state-of-the-art methods in terms of accuracy metric. By finetuning the corresponding classification heads with a small amount of data on the pretrained UniEEG Encoder, models adapted to different tasks and datasets can be realized. With the electrode-wise time-frequency pretraining, the UniEEG obtains universal EEG representation, and significantly improves the capability and generalization of the model. For example, on Neuro-Marketing Dataset Yadava et al. (2017) (the third column) for image-based classification task, the outperforms prior art Spampinato et al. (2017) by 13.71%.

We also conduct experiments of UniEEG in task-specific paradigm, where we train and evaluate the model in each dataset independently. Experimental results are shown in the "Ours w/o pretraining" (third row) of Tab. 1. We observe that the removal of pretraining in UniEEG decreases the performance by a large margin (i.e., 6.27% in DEAP). This further demonstrates that the electrode-wise pretraining-finetuning paradigm of EEG tasks outperform previous task-specific paradigm, indicating the superiority of UniEEG.

It should be noted that previous state-of-the-art methods (i.e., Bazgir et al. (2018)) would take other modalities (i.e., electro-oculogram, facial videos) as input and thus get a good performance, while we only report the results that takes only EEG signals as input for fairness. Moreover, there are different evaluation strategies of EEG tasks, including holdout validation, K-fold Cross-Validation, leave-one-subject-out validation and so on. In Tab. 1, the results that evaluated with holdout validation are reported.

### 5.3 ABLATION STUDIES

In this section, we conduct a comprehensive ablation study to analyse various aspects of  design.

### 5.3.1 IMPACT OF SIGNAL DOMAIN

In our implementation, we leverage the time-frequency data of EEG, which contains both the temporal and spectral information. To investigate the effect of data domain on . we conduct experiments on the model based solely on time domain or frequency domain. In fairness, each single domain data is also transformed to an image by repeating the other axis. For example, for time domain data with size of $T \times 1$, we repeat the whole data for $F$ times, resulting an image with size of $F \times T \times 1$ The results are shown in Tab. 4. We observe the absence of each domain leads to the decrease of performance. For example, compared with training with time-frequency domain data,

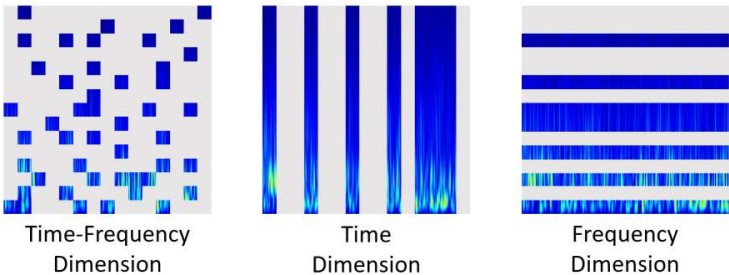

Figure 3: Different masking strategies. Left: masking on the time-frequency dimension. Middle: masking on the time dimension. Right: masking on the frequency dimension.

Table 4: Comparison on different signal domains.

| Method | Ger+Aus | DEAP |
|---|---|---|
| Time Only | 20.17% | 81.18% |
| Frequency Only | 17.41% | 63.50% |
| Time-Frequency | **22.59**% | **85.61**% |

Table 5: Comparison with different reconstruction targets.

| Target | Ger+Aus | DEAP |
|---|---|---|
| PCA | 20.15% | 80.54% |
| dVAE token | 22.15% | 78.76% |
| Energy | **22.59**% | **85.61**% |

the accuracy of Deap decreases by 4.43% when using time-only domain data. This suggests the cues of the cross-domain data help regularize the signal representation and improve the final performance of subsequent task.

### 5.3.2 DECODER DESIGN

In our , the decoder is designed to reconstruct the original signal from the encoded unmasked signal patches and added mask tokens. Here we conduct ablation study for the decoder design on different settings. Intuitively, such a decoder has a limited impact on downstream tasks, where the decoder is replaced by a classifier. Tab. 2 shows the comparisons between different depths of the decoder (number of transformer blocks). Tab. 3 shows the comparisons between different decoder widths (the hidden dimension of the transformer layers). We can see that the change in decoder settings have limited influence on the classification performance, which we reason with the unfrozen parameters of UniEEG-encoder in the finetuning process.

To investigate the representational ability of , we also conduct experiments with a frozen encoder, results shown in the last two column of Tab. 2 and Tab. 3. We observe that the difference of decoder depths or widths will greatly influence the performance of downstream task. For instance, the 8 transformer layers can improve the final accuracy in Ger+Aus task by 7.62%, compared with 1 transformer layer. This indicates that the different design of the UniEEG-decoder would change the representational space of the UniEEG-encoder, which yields different performances when such representation is frozen. However, compared with the unfrozen encoder, the frozen setting is suboptimal, which has a decrease of around 2.81% performance. Thus in other experiments we do not freeze the encoder parameters to get an optimal performance.

### 5.3.3 RECONSTRUCTION TARGET

Tab. 5 shows the comparisons between different reconstruction targets. In previous experiments, the reconstruction period are mainly based on pixels, similar to visual images. In this study, following He et al. (2022), we replace the reconstruction target from Time-Frequency Signal to PCA in the patch space and dVAE, results shown in Tab. 5. We observe that both of the replacements decrease the performance. The potential reason is that the naive setting that reconstructs the signal directly allows the model to capture more general features, which benefits the downstream classification tasks.

### 5.3.4 MASKING STRATEGY

In our , we mask the time-frequency EEG signals in the time-frequency domain along both time and frequency dimensions, the same as the traditional spatial mask of an image. Here we conduct



Figure 4: Visualization of reconstruction results. Left: raw EEG signal. Middle: masked EEG signal. Right: reconstruction results.

Table 6: Impact of keeping or removing the mask tokens from the encoder input.

| Mask Token | Ger+Aus | DEAP |
|---|---|---|
| encoder w/ [M] | 20.49% | 84.56% |
| encoder w/o [M] | **22.59%** | **85.61%** |

Table 7: Comparison with different masking strategies.

| Masking Strategy | Ger+Aus | DEAP |
|---|---|---|
| Time-Frequency | **22.59%** | **85.61%** |
| Time | 20.79% | 81.18% |
| Frequency | 14.02% | 66.17% |

experiments to compare different mask strategies. As illustrated in Fig. 3, we perform experiments on three types of masking strategies, including masking along the time dimension, along the frequency dimension, and along both the time and frequency dimensions. Tab. 7 shows the results. We can see that the best performance is achieved when we mask on the time-frequency dimensions, which yields 1.80% improvement than masking on the time dimension and 8.57% improvement than masking on the frequency dimension in Ger+Aus.

### 5.3.5 MASK TOKENS

In our , we remove masked signal patches during the encoding process, while during the decoding process, mask tokens are added at the masking place to indicate the presence of a missing patch to be predicted. Here we conduct experiments on mask token design. As shown in Tab. 6, the encoder with mask tokens decreases the overall performance by 2.10% in Ger+Aus and 1.05% in DEAP. The probable reason is that the added masks in the encoder is shared and do not exist in the original signal, which degrades the performance.

### 5.3.6 PATCH SIZE

In previous experiments, the patch size of the signal token is 25. In this study, we investigate the impact of different patch sizes. As shown in Tab. 8, increasing the patch size of the time-frequency "image" would improve the final results, but too big patch would cause a collapse of the performance.

### 5.3.7 FREQUENCY RANGE

In previous experiments, the frequency range of EEG signal is limited to between 1Hz and 49Hz. In this study, we conduct experiments on the different frequency range of EEG signal. We follow Luo et al. (2023) and split the with five basic brain waves: $\delta$ wave, $\theta$ wave, $\alpha$ wave, $\beta$ wave and $\gamma$ wave, where the frequency ranges are 1-4 Hz, 4-8 Hz, 8-12 Hz, 12-27 Hz and 27-49 Hz respectively, results shown in Fig. 9. We can see that $\alpha$ wave is good at recognizing image (Ger+Aus) and $\theta$ wave is good at emotion analysis (DEAP), but they all underperform that using all frequencies.

### 5.3.8 SIGNAL CROPPED SIZE

During pretraining period, the signal cropped size varies in different datasets. To align this, we randomly crop and resize them to a fixed length $t = 100$. Here we conduct experiments on the impact of cropped size, results shown in Tab. 5. We can see that the computation cost increases steadily as the signal cropped size increases, while the performance begins to decrease after reaching its peak at $t = 100$. The reason is the effective duration of EEG activity is relative stable. Too short

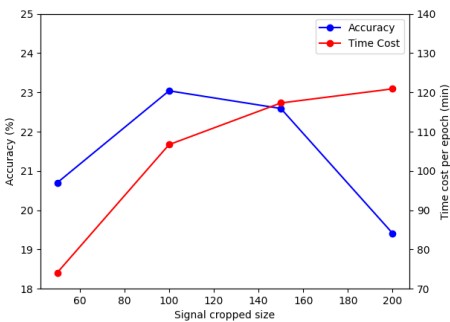

Figure 5: Impact of signal cropped size on performance and computation cost. The computation cost increases steadily as the signal cropped size increases, while the performance begins to decrease after reaching its peak at $t = 100$.

Table 8: Comparison with different signal patch sizes.

| Patch Size | Ger+Aus | DEAP |
|---|---|---|
| 5 | 20.70% | 80.79% |
| 10 | **23.04**% | 83.18% |
| 25 | 22.59% | **85.61**% |
| 50 | 19.41% | 76.41% |

Table 9: Comparison with different frequency ranges.

| Frequency Range | Ger+Aus | DEAP |
|---|---|---|
| $\delta$ (1-4) | 18.40% | 73.74% |
| $\theta$ (4-8) | 17.59% | **85.43**% |
| $\alpha$ (8-12) | **20.38**% | 83.74% |
| $\beta$ (12-30) | 17.56% | 74.19% |
| $\gamma$ (30-49) | 19.49% | 77.62% |
| ALL (1-49) | **22.59**% | **85.61**% |

cropped length leads to missing valid information, while too long cropped length leads to redundant information.

### 5.3.9 CHANNEL AGGREGATION FUNCTION

We utilize the 1D convolution as a learnable aggregation function to fuse the features from different channels. When finetuning, we flatten all of the EEG features and input them to the 1D convolution. In this study, we investigate the effects of other aggregation functions, all of which achieve good performance, shown in Tab. 10. This shows that simply aggregating the features of these single channels with the pretrained UniEEG encoder can achieve good results, indicating the flexibility of the proposed UniEEG.

Table 10: Comparison with different channel aggregation functions.

| Method | Ger+Aus | DEAP |
|---|---|---|
| 1D Convolution | 22.59% | **95.61**% |
| Fully Connected | **23.04**% | 95.16% |
| Mean Pooling | 22.18% | 94.87% |

## 6 CONCLUSION

In conclusion, we presents UniEEG, the first electrode-wise time-frequency pretraining model for EEG. During pretraining stage, we divide the electrode channels into individual channel and employ an encoder-decoder structure to model and reconstruct the time-frequency signals. In finetuning phase, we exploit an aggregation module to fuse the multi-channel information, enabling the model to perform diverse downstream tasks. Extensive experiments on different tasks demonstrate the effectiveness and generalizability of our proposed architecture, highlighting the potential of our approach. Overall, our findings establish the value and versatility of UniEEG as a pretraining model for EEG analysis, offering promising prospects for advancing our understanding and utilization of EEG signals in diverse domains.

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

## A  APPENDIX

## B  EEG DATA COLLECTION

The classic EEG datasets Koelstra et al. (2012); Zheng & Lu (2015); Daly et al. (2019); Chavarriaga & Millán (2010); Brunner et al. (2008); Steyrl et al. (2014); Leeb et al. (2008); Cho et al. (2017); Schalk et al. (2004); Luciw et al. (2014); Kaya et al. (2018); Schirrmeister et al. (2017b); Bhatt (2012); Dornhege et al. (2004); Gifford et al. (2022); Grootswagers et al. (2022); Spampinato et al. (2017); Nguyen et al. (2017) we collect cover 6 tasks, which include: 1) **Sentiment Analysis**: using EEG data to identify and evaluate the emotional state of an individual; 2) **Music Imagery**: studying and analyzing the electrical activity of the brain while a person imagines or mentally processes music; 3) **Event-Related Potential (ERP)**: analyzing the brain's electrical activity in response to specific events or stimuli, such as visual, auditory, or sensory stimuli. 4) **Motor Imagery**: referring to the mental simulation or visualization of specific motor movements or actions without physically performing them. 5) **Image-based EEG Classification**: using EEG data to classify images or other visual stimuli. 6) **Speech Imagery Classification**: using EEG data to categorize or classify different aspects of speech without physically hearing them.

Below we introduce each data set in detail. **SEED**Zheng & Lu (2015): The SJTU Emotion EEG Dataset (SEED), a comprehensive compilation of EEG datasets, is a significant contribution from the BCMI laboratory, under the expert guidance of Prof. Bao-Liang Lu. This dataset derives its name from its initial version, which primarily focused on emotion-related EEG data. However, in its current form, SEED has expanded its scope beyond just emotional data to include a vigilance dataset as well, thereby enhancing its utility for a broader range of neurological and psychological research.

**Neuromarketing**Yadava et al. (2017): The Neuromarketing dataset, designed to decipher consumer preferences and predict behavior for optimal product utilization, encompasses a detailed collection of EEG signals. These signals are meticulously recorded from 25 participants, aged between 18 to 38 years. Participants engage in viewing a curated selection of consumer products on a computer screen, with the EEG data captured using all 14 channels. The dataset focuses on a set of 14 distinct products, each presented in three variants, culminating in a total of 42 (14 × 3) unique product images. This leads to an extensive dataset of 1050 (42 images × 25 participants) EEG recordings. Accompanying each image viewing, participants' feedback is solicited in the form of like/dislike responses. Each product image is displayed for a duration of 4 seconds, during which EEG signals are simultaneously recorded. Following the display of each image, participants' choice preferences are meticulously documented. To ensure authenticity and accuracy, participants are instructed to provide genuine responses regarding their product preferences throughout the data collection process.

**DEAP** Koelstra et al. (2012): The DEAP dataset is a comprehensive multimodal resource designed for studying human affective states. This unique dataset includes electroencephalogram (EEG) and peripheral physiological signal recordings from 32 participants, who are engaged in watching 40 one-minute long excerpts of various music videos. These participants provide subjective ratings for each video, assessing them on a scale of arousal, valence, like/dislike, dominance, and familiarity. Enhancing the depth of this dataset, frontal face videos are also captured for 22 out of the 32 participants, offering an additional dimension of emotional response analysis. The selection of stimuli for this dataset is conducted using a novel approach.

**MIBCI** Cho et al. (2017): The dataset described in this survey is a comprehensive resource for studying motor imagery brain-computer interface (MI BCI) research. It not only includes EEG datasets from 52 subjects but also incorporates various additional data types and metadata. The EEG datasets provide essential information for determining statistical significance and are further

categorized into well-discriminated datasets (38 subjects) and less-discriminative datasets. This categorization offers researchers the opportunity to explore human factors that contribute to variations in MI BCI performance. The inclusion of additional data such as results from psychological and physiological questionnaires, EMG datasets, 3D EEG electrode locations, and EEG recordings during non-task related states enhances the dataset's richness. The availability of metadata, including the questionnaire responses, EEG coordinates, and EEGs for non-task related states, opens avenues for subject-to-subject transfer and facilitates investigations into various aspects related to MI BCI performance. Researchers can leverage these resources to explore human factors and their impact on MI BCI, ultimately advancing the field and potentially improving the transferability of MI BCI systems.

**Grasp and Lift** Kaggle (2021): The Grasp and Lift dataset is a rich and multifaceted resource primarily focused on electroencephalogram (EEG) data for motor imagery (MI) brain-computer interface (BCI) research, encompassing a diverse array of data from 52 subjects. This dataset not only includes EEG recordings during MI tasks but also offers valuable supplementary information, such as results from psychological and physiological questionnaires, electromyogram (EMG) data, and precise locations of 3D EEG electrodes. Additionally, EEG recordings during non-task related states are provided, offering a comprehensive view of the subjects' brain activity. A distinctive feature of this dataset is its meticulous validation process. It employs methods like the analysis of the percentage of bad trials, event-related desynchronization/synchronization (ERD/ERS), and classification analysis to ensure data quality. The dataset demonstrates typical MI patterns, such as contralateral ERD and ipsilateral ERS in the somatosensory area. Notably, a significant portion of the dataset (73.08The dataset is categorized into well-discriminated and less-discriminative datasets based on the clarity and distinctiveness of the EEG signals. This classification provides a unique opportunity for researchers to investigate various human factors influencing MI BCI performance and explore subject-to-subject transfer methodologies. The inclusion of comprehensive metadata, such as questionnaire responses, EEG coordinates, and EEGs for non-task states, further enhances the dataset's utility for diverse research applications in the field of BCI.

**EEG Motor Imagery** Kaya et al. (2018): This dataset features over 1500 one- and two-minute EEG recordings from 109 volunteers, using the BCI2000 system. It focuses on motor/imagery tasks across 14 experimental runs per subject, including two baseline runs (one with eyes open, one closed) and three runs for each of four tasks: (1) Physical fist movement when a target appears on the screen, (2) Imagined fist movement for a similar target, (3) Physical movement of fists or feet depending on the target's position, and (4) Imagined movement of fists or feet for corresponding targets. This dataset is ideal for brain-computer interface research, exploring physical and imagined motor activities.

**BCI Competition III/IV** Dornhege et al. (2004); Brunner et al. (2008); Leeb et al. (2008): The 'BCI Competition III/IV' is designed to evaluate signal processing and classification methods in Brain-Computer Interface (BCI) research. Focused on motor imagery, especially in the context of sports, it offers a comprehensive challenge with multiple motor imagery paradigms. This dataset serves as a crucial benchmark for advancing BCI technology.

**Aus** Gifford et al. (2022): The Aus dataset is a significant contribution to the study of the neural basis of object recognition and semantic knowledge. This dataset includes electroencephalography (EEG) responses from 50 subjects to 1,854 object concepts, represented through 22,248 images from the THINGS stimulus set, a specially designed high-quality image database for human vision research. THINGS-EEG offers neuroimaging data correlated with a vast array of objects and concepts, facilitating extensive research in visual object processing in the human brain.

**Ger** Grootswagers et al. (2022): The Ger dataset provides a comprehensive collection of high temporal resolution EEG responses to object images on natural backgrounds, crucial for understanding the rapid transformations in visual object recognition by the human brain. It comprises data from 10 participants across 82,160 trials, covering 16,740 image conditions.

**EEG-Based Visual** Spampinato et al. (2017): The EEG-Based Visual dataset contains EEG data recorded from six subjects (five male, one female) while they were shown visual stimuli of objects. These subjects were selected for their homogeneity in age, education, and cultural background and screened by a professional physicist to ensure no interfering conditions. The visual stimuli comprised 2,000 images from 40 classes in a subset of ImageNet, each shown for 0.5 seconds in 25-

second bursts, followed by a 10-second pause. The experiment, lasting 23 minutes and 20 seconds, used a 128-channel EEG cap with active electrodes and high-resolution data acquisition at 1000 Hz. The EEG data focuses on the Beta and Gamma frequency bands, relevant to cognitive processes in visual perception. The first 40 ms of each EEG sequence were discarded to avoid interference from previous images, with the subsequent 440 ms used for analysis. This resulted in 12,000 EEG sequences, offering a detailed exploration of cognitive processing in visual object recognition.

**Speech Imagery** Nguyen et al. (2017): This S is part of a study investigating the use of imagined speech for brain-computer interface (BCI) applications. It includes EEG signals collected from 15 subjects, focusing on the imagined pronunciation of vowels, short words, and long words. It is an important benchmark of speech imagery.

## C  DATA PREPROCESS

The primary challenge in preprocessing large-scale EEG signals lies in the variations of collection parameters such as sampling frequency and the number of electrodes across different datasets, each adhering to its unique collection paradigm. To address this, we employ two primary strategies: aligning the sampling frequency and standardizing the number of channels.

Firstly, to standardize the sampling frequency, we adjust all EEG data to a uniform rate of 100Hz. This involves either upsampling or downsampling the signals. Upsampling is achieved through linear interpolation, which estimates intermediate values, while downsampling utilizes a uniform sampling method that selects consistent intervals. Following this frequency alignment, we transform the time-domain EEG signals into the time-frequency domain using the Continuous Wavelet Transform (CWT). This transformation facilitates a more nuanced analysis of the signals. We further refine the data by applying a simple filter, eliminating frequencies below 2Hz and above 50Hz to focus on the most relevant signal range.

Secondly, to manage the variation in the number of electrodes (channels), we introduce an electrode-wise pretraining and fine-tuning approach. Acknowledging that EEG signals can be represented uniformly despite channel differences, we treat each channel as an independent sample. This strategy allows us to handle datasets with varying channel numbers effectively. Additionally, we align the data collection time across different paradigms by employing techniques similar to image data augmentation. Specifically, we randomly crop and resize the EEG signal along the time dimension, ensuring consistency in signal length.

It's important to note that during downstream tasks, the data from different channels are not treated separately but are instead integrated through a fusion operation. Furthermore, the data collection time is resized to a pre-set dimension only during the pretraining period.

In our methodology, we consciously avoid employing other complex preprocessing methods to minimize information loss and maintain the integrity of the EEG data, ensuring that the processed signals remain as representative and accurate as possible of the original recordings.

