# OpenReview forum: "UniEEG: Advancing Universal EEG Representation with Electrode-Wise Time-Frequency Pretraining"
_ICLR.cc/2025/Conference — ICLR 2025 Conference Withdrawn Submission_

### Official Review · Reviewer_Pegs · 2024-10-21

**Soundness:** 1
**Presentation:** 2
**Contribution:** 1
**Rating:** 1
**Confidence:** 4

**Summary:**

This article describes a method for constructing a large-scale EEG model, including electrode-wise techniques to unify representations across different datasets, a time-frequency encoder, and a mask-based pretraining method for reconstruction.

**Strengths:**

1.The topic is of practical significance.
2.The diagrams are clear.

**Weaknesses:**

1.There is a significant issue with the claim of innovation: The electrode-wise preprocessing and mask-based mutual reconstruction methods proposed for constructing the large-scale EEG model are very similar to those in LaBraM[1]. However, the article claims these methods are newly proposed without citing this reference. Notably, this reference is a spotlight paper of ICLR 2024 and is well-known in the field. The lack of corresponding research is unacceptable.
2.Lack of technical depth: For a scientific paper, the absence of any equations throughout the text is problematic. Simple descriptions fail to provide detailed explanations. Although the use of time-frequency features is mentioned, the methods of extracting and unifying features from different datasets are not thoroughly described.
3.Unconvincing experimental results: Only two baseline comparisons are provided, one from 2018 and another from 2015, without comparisons to any recent models in the EEG field, including LaBraM.
4.Formatting issues: The font in Table 1 is significantly reduced, making it difficult to read; the line spacing for headings in Sections 5.3.2 and 5.3.7 is reduced, even overlapping with the text, which does not meet ICLR standards.

[1]Wei-Bang Jiang, Li-Ming Zhao, and Bao-Liang Lu. Large brain model for learning generic representations with tremendous eeg data in bci. arXiv preprint arXiv:2405.18765, 2024.

**Questions:**

1.How are the frequency domain features extracted from different datasets, and are there any specific challenges in unifying these features across different datasets?

---

### Official Review · Reviewer_unkY · 2024-10-27

**Soundness:** 2
**Presentation:** 1
**Contribution:** 2
**Rating:** 3
**Confidence:** 5

**Summary:**

The paper introduces UniEEG, an electrode-wise time-frequency pretraining model for EEG signal processing. By utilizing the Continuous Wavelet Transform (CWT), UniEEG captures time-frequency features, making it robust to signal noise and delays commonly found in EEG data. UniEEG addresses the limitations of previous task-specific models by leveraging data from nearly 20 publicly available datasets spanning six EEG tasks. The proposed model employs a self-supervised Masked Autoencoder (MAE) framework to pre-train on time-frequency EEG data and fine-tune on downstream tasks. While the model offers promising results, several weaknesses, including writing issues, missing comparisons with key baselines, and limited novelty, detract from the overall quality of the submission.

**Strengths:**

* UniEEG employs Continuous Wavelet Transform (CWT) to capture the time-frequency characteristics of EEG signals.
* The paper presents extensive ablation studies, which provide insight into key design choices like masking strategies, decoder depths, and the impact of signal domain (time vs. frequency).

**Weaknesses:**

* Poor Writing quality. The manuscript suffers from numerous writing and formatting issues. For instance, there are two "Introduction" sections on Page 1. Section 4.1.2 contains an incomplete sentence: “most lengths It should be” is missing a period. Similarly, Section 5.3.1 has an incomplete sentence: “To investigate the effect of data domain on.” Additionally, Section 5.3.2 contains “To investigate the representational ability of,” and in Section 5.3.4, “In our” is left hanging. These errors make the paper difficult to follow and detract from its overall quality.
* Insufficient related work. The authors claim in Section 3.2 that "there are no studies validating MSM in EEG signals, which is the main focus of our work." However, several recent studies have already explored the use of masked signal modeling (MSM) for EEG, such as [1], [2], and [3]. These works should be discussed and compared to clarify how UniEEG advances the field.
* Lack of baseline comparison. The abstract and introduction have significant overlap with LaBraM [1], a pioneering foundation model for EEG signals. However, UniEEG does not include a comparison with LaBraM or other commonly used models like BIOT [4]. Instead, the paper only compares UniEEG against its own variations and two out-of-date baselines, which is insufficient to demonstrate its advantages or novelty.
* Limited novelty. The approach of UniEEG is largely a straightforward application of vanilla MAE (Masked Autoencoder) on EEG data. There isn’t a significant methodological innovation beyond adapting a pre-existing model to EEG.
* Missing implementation details. Essential details, such as the training, validation, and test split, are absent. This information is critical for replication and for assessing the reliability of the reported results.

Overall, due to the poor writing, limited novelty, insufficient related work, and lack of robust comparisons, this paper falls short of the standards for a top-tier ML conference like ICLR. However, the ablation studies show some interesting findings. I suggest the authors make substantial revisions and consider submitting the work to a journal instead.

[1] Large Brain Model for Learning Generic Representations with Tremendous EEG Data in BCI. ICLR 2024.

[2] EEG2Rep: Enhancing Self-supervised EEG Representation Through Informative Masked Inputs. KDD 2024.

[3] Neuro-BERT: Rethinking Masked Autoencoding for Self-Supervised Neurological Pretraining. IEEE Journal of Biomedical and Health Informatics 2024.

[4] BIOT: Biosignal Transformer for Cross-data Learning in the Wild. NeuIPS 2023.

**Questions:**

In the current setup, each electrode is modeled independently, and representations are fused after the encoder. This late fusion approach may miss out on capturing important temporal-frequency correlations between electrodes during pre-training. Have you considered a joint encoding approach where all channels (electrodes) are combined into an F×S×C signal, where C is the number of channels? Spatial embeddings could be added to distinguish the electrodes, allowing interactions between time, frequency, and space. It would be interesting to see if this improves performance.

---

### Official Review · Reviewer_PYXu · 2024-10-31

**Soundness:** 3
**Presentation:** 1
**Contribution:** 2
**Rating:** 3
**Confidence:** 4

**Summary:**

The paper proposed  an universal EEG pre-training method that can integrate datasets with different numbers of electrodes for training. The evaluation results demonstrated that the proposed method can out-perform the SOTA methods that were trained on individual datasets.

**Strengths:**

1. The figures have high quality and are informative.
2. The identified challenge is a key gap in the literature that attracted a lot of research interest.

**Weaknesses:**

1. The general writing quality can be improved a lot. For example, there are two introduction sections, missing spaces in section1,  4.1.1, sentence starting with 'And' in section 5.1.1, grammatical errors in section 5.2, extra spaces before period and missing words in section 5.3.1, 5.3.2, 5.3.4, format error in section 5.3.2.
2.  The paper claimed to be 'the first electrode wise time-frequency pretraining model' which unfortunately not the case. Please check the following references:
Yang, C., Westover, M., & Sun, J. (2024). Biot: Biosignal transformer for cross-data learning in the wild. Advances in Neural Information Processing Systems, 36.
Yi, K., Wang, Y., Ren, K., & Li, D. (2024). Learning topology-agnostic eeg representations with geometry-aware modeling. Advances in Neural Information Processing Systems, 36.
3. The 'resize' operation for segments with random duration is unclear. Is it done through padding or re-sampling? If so, does it mean that the input segments have different sampling rate?
4. Table 1 is too small and unreadable.
5. It is unclear if the SOTA results were obtained through re-implementation and experiments under the same setup or simple report of their performance from their original paper. The s.t.d. of the performance metrices should be provided and significance figures should be reported when making comparison. How were the holdout validation set designed?
6. The data split for evaluation is unclear. Section 5.1.1 reported that there were 18 datasets used in total, 16 were used for pre-training and 12 were used for evaluation. Were the datasets split based on the subject's identity or sessions? This part need more clarification.

**Questions:**

1. It is unclear how does the proposed method learn the functional connectivity between the channels from different brain regions since the pre-training was performed channel by channel unlike Yi et al. (2024). Can the authors clarify this?
2. How does the masking percentage effect pre-training performance?

---

### Official Review · Reviewer_U6nw · 2024-11-06

**Soundness:** 1
**Presentation:** 1
**Contribution:** 1
**Rating:** 1
**Confidence:** 5

**Summary:**

This submission lacks clarity and coherence, making it challenging to engage with. The reviewer has identified several ambiguous statements suggesting that the authors may not possess a solid understanding of EEG signals and various BCI tasks. The application of CWT to EEG is neither innovative nor distinctive. Additionally, the filtering of EEG signals within the 2-50 Hz range may limit the applicability of the approach to only motor imagery (MI) or steady-state BCIs, thus rendering the concept non-universal. Furthermore, the submission appears to be highly philosophical while lacking the necessary details to ensure research reproducibility. Consequently, the reviewer is left with no option but to recommend rejection.

**Strengths:**

Authors are strongly encouraged to thoroughly investigate BCI and EEG issues prior to proposing any universal solutions. Moreover, enhancing the clarity of manuscript writing is imperative to ensure that readers can comprehend and replicate the research findings. Importing machine learning methodologies from other domains, such as NLP, into EEG applications poses significant risks and concerns.

**Weaknesses:**

The submission presents significant challenges in readability and comprehension. It appears that the authors may lack an understanding of EEG processing issues and propose generalized solutions that lack clarity and relevance.

**Questions:**

The authors' failure to investigate EEG and BCI challenges prior to proposing "a universal solution" raises significant concerns, particularly given that such a solution lacks coherence across different BCI paradigms. The use of a simplistic signal resampling method and standard continuous wavelet transform (CWT) application, following dubious bandpass filtering and neglecting to address artifacts, is fundamentally flawed. Why did the authors neglect to familiarize themselves with the field of BCI prior to proposing universal solutions?

---

### Note · Authors · 2024-11-15

I have read and agree with the venue's withdrawal policy on behalf of myself and my co-authors.